

# The Permafrost and Organic LayEr module for Forest Models (POLE-FM) 1.0

Winslow D. Hansen[1*], Adrianna Foster[3], Benjamin Gaglioti[4], Rupert Seidl[2,5], Werner Rammer[2]

[1]Cary Institute of Ecosystem Studies, Millbrook, NY, USA, 12545
[2] Technical University of Munich, School of Life Sciences, 85354 Freising, Germany
[3] National Center for Atmospheric Research, Boulder, CO, USA, 80035
[4]Water and Environmental Research Center, Institute of Northern Engineering, University of Alaska Fairbanks, Fairbanks, AK USA, 99775
[5] Berchtesgaden National Park, 83471 Berchtesgaden, Germany

*Correspondence to*: Winslow D. Hansen (hansenw@caryinstitute.org)

**Abstract.** Climate change and increased fire are eroding the resilience of boreal forests. This is problematic because boreal vegetation and the cold soils underneath store approximately 30% of all terrestrial carbon. Society urgently needs projections of where, when, and why boreal forests are likely to change. Permafrost (i.e., subsurface material that remains frozen for at least two consecutive years) and the thick soil-surface organic layers (SOLs) that insulate permafrost are important controls of boreal forest dynamics and carbon cycling. However, both are rarely included in process-based vegetation models used to simulate future ecosystem trajectories. To address this challenge, we developed a computationally efficient permafrost and SOL module named the Permafrost and Organic LayEr module for Forest Models (POLE-FM) that operates at fine spatial (1 ha) and temporal (daily) resolutions. The module mechanistically simulates daily changes in depth to permafrost, annual SOL accumulation, and their complex effects on boreal forest structure and functions. We coupled the module to an established forest landscape model, iLand, and benchmarked the model in interior Alaska at spatial scales of stands (1 ha) to landscapes (61,000 ha) and over temporal scales of days to centuries. The coupled model could generate intra- and inter-annual patterns of snow accumulation and active layer depth (portion of soil column that thaws throughout the year) consistent with independent observations in 17 instrumented forest stands. The model was also skilled at representing the distribution of near-surface permafrost presence in a topographically complex landscape. We simulated 34.6% of forested area in the landscape as underlain by permafrost; a close match to the estimated 33.4% from the benchmarking product. We further determined that the model could accurately simulate moss biomass, SOL accumulation, fire activity, tree-species composition, and stand structure at the landscape scale. Modular and flexible representations of key biophysical processes that underpin 21[st]-century ecological change are an essential next step in vegetation simulation to reduce uncertainty in future projections and to support innovative environmental decision making. We show that coupling a new permafrost and SOL module to an existing forest landscape model increases the model's utility for projecting forest futures at high latitudes. Process-based models that represent relevant dynamics will catalyze opportunities to address previously intractable questions about boreal forest resilience, biogeochemical cycling, and feedbacks to regional and global climate.



## 1 Introduction

The boreal forest is warming at a rate at least twice the global average (IPCC, 2021; Chylek et al., 2022) causing
climate-sensitive disturbances, like forest fire, to increase (Seidl et al., 2020; Walker et al., 2020). Together, pronounced
warming and larger, more severe fires are initiating abrupt changes in forest cover, structure, functions, and tree-species
composition (Johnstone et al., 2010; Alexander and Mack, 2016; Walker et al., 2019; Mack et al., 2021; Baltzer et al., 2021);
trends that will likely continue for at least the next several decades (Mekonnen et al., 2019; Foster et al., 2019, 2022). This is
important because biophysical properties of the boreal forest underpin feedbacks to regional climate (Foley et al., 1994; Chapin
et al., 2008; Rogers et al., 2013; Potter et al., 2020), and ~30% percent of all terrestrial organic carbon stocks are stored in the
biome (Lorenz and Lal, 2010; Schurr et al., 2018). Some portion of those stocks could be released to the atmosphere and
further accelerate warming (Anderegg et al., 2022). Thus, society urgently needs projections of where, when, and why the
boreal forest will change.

Ecological legacies are the organismal adaptations (i.e., information), physical materials, and energy that persist in
ecosystems through multiple disturbances (Ogle et al., 2015). Legacies will underpin how the boreal forest responds to climate
change and fire (Turetsky et al., 2016; Johnstone et al., 2016). For example, adaptive traits, like cone serotiny (cones that stay
closed for many years until heated by fire) and asexual resprouting, are information legacies that facilitate postfire forest
recovery (Johnstone et al., 2009, 2010). Thick moss-dominated soil-surface organic layers (SOL) form over decades of postfire
forest development, and a portion often escapes burning in the subsequent fire, leading to accumulation of SOL over multiple
fire cycles (Walker et al., 2018). This serves as a physical legacy that preserves permafrost (subsurface material that remains
frozen for at least two consecutive years) (Kasischke and Johnstone, 2005; Jorgenson et al., 2010) and shapes tree species
composition by controlling seedling germination and establishment (Johnstone et al., 2020). In conjunction with insulative
physical legacies, energy legacies of past temperature regimes also maintain permafrost underneath forests where current air
temperature would otherwise not support it (Schuur and Mack, 2018).

Physical and energy legacies underpin spatio-temporal patterns of permafrost at multiple scales. At the biome scale,
permafrost is continuous in the north, becomes discontinuous, sporadic, and is then eventually absent from the southern boreal
forest (Obu et al., 2019). Within the discontinuous zone, the permafrost distribution is heterogeneous, varying on fine spatial
scales with topography, dominant forest type, and fire history (Brown et al., 2016; Gibson et al., 2018). Permafrost dynamics
are particularly important for shaping boreal forest structure and function (Turetsky et al., 2010; Baltzer et al., 2014; Dearborn
and Baltzer, 2021). Within permafrost-affected soils, a portion of the soil column termed the "active layer" undergoes an
annual cycle of freezing and thawing. The annual maximum active-layer depth can vary from a few centimeters to several
meters (Smith et al., 2022). This freezing and thawing determines the seasonality, vertical distribution, and amount of plant-
available soil water and influences nutrient availability (Abbott and Jones, 2015; Young-Robertson et al., 2017).

In response to continued warming, annual maximum active layer depth is predicted to increase, and the distribution
of permafrost will likely contract, with large hydrologic and biogeochemical consequences (Pastick et al., 2015; Schuur and



Mack, 2018). Increasing wildfire (Veraverbeke et al. 2017, Phillips et al. 2022) will also impact permafrost by combusting SOLs and altering forest regeneration pathways (Baltzer et al. 2021, Johnstone et al. 2010). However, permafrost and the legacies that affect its dynamics, are rarely considered in forest models. In fact, just a handful of models explicitly simulate permafrost (Foster et al., 2019; Gustafson et al., 2020; Kruse et al., 2022), and those that do often operate at relatively coarse

spatial (≥ 25 ha grid cells) and/or temporal (≥ monthly) resolutions (but see Kruse et al. 2022). This makes it difficult to capture the fine-scale spatial heterogeneity of permafrost distributions and the effects of daily temperature variability on plant water availability during short, but critical shoulder seasons. Further, most existing permafrost algorithms rely on computationally intensive numerical methods (e.g., Sitch et al. 2003, Beer et al. 2007), limiting the spatio-temporal resolutions at which they can be applied, particularly across broad domains.

75        To address this challenge, we present the Permafrost and Organic LayEr module for Forest Models (POLE-FM) that was designed to mechanistically simulate daily changes in active layer depth, annual SOL accumulation, and the associated ecological effects on boreal forests and fire at a fine spatial resolution (i.e., grain of ~1-ha) in a computationally efficient manner (Fig. 1). When paired with a state-of-the-art forest model, such as iLand, the module allows for simulation of complex feedbacks among forests, fire, and permafrost dynamics in topographically complex landscapes under historical and future

conditions. In this paper, we describe the module and benchmark its ability to represent permafrost and SOLs in forest stands to landscapes of interior Alaska across days to centuries.

## 2 Model description

### 2.1 Permafrost and SOL module

        The module represents daily changes in active layer depth and long-term trends (years to decades) in permafrost

presence. Permafrost is represented based on physical principles of heat transport through vegetation and soil media with varying thermal resistances affected by soil moisture content. We incorporate the insulating effects of snow and deep SOLs and capture transient shifts between permafrost regimes (e.g., a transition from temporally continuous to sporadic permafrost due to climate change). Moreover, we aimed for a computationally efficient approach that operates well within the runtime and memory constraints of forest models. The module tracks the energy fluxes that thaw and freeze water at the edge of the

active layer (zero isoline, or the depth at which soil temperature is 0 °C), requires only a few state variables, and provides daily values of active layer depth with little computational overhead by avoiding iterative numerical approximations of differential equations.

        To capture daily changes in active layer depth, we first estimate the thermal resistances $R$ [m² W⁻¹ K⁻¹] of snow (when present), SOL, and the mineral soil layer (Eq. 1).

$$R = \frac{Snow.Depth}{Snow.k} + \frac{SOL.Depth}{SOL.k} + \frac{M.Soil.Depth}{M.Soil.k},$$    (1)





Snow depth is represented as a function of the precipitation that falls during days with mean air temperature below 0°C and the density of snow (set at 190 kg m⁻³) (Bonan, 1991; Bennett et al., 2019). We set snow thermal conductivity, $Snow.k$, at 0.3 w m⁻¹ K⁻¹ (Cook et al., 2008). SOL depth is estimated based on the mass of live and dead mosses and litter
pools in each grid cell. SOL thermal conductivity, $SOL.k$ is set at 0.09 w m⁻¹ K⁻¹ (Hinzman et al., 1991; O'Donnell et al., 2009).

Characteristics of the mineral soil layer that determine its conductivity are explicitly considered. We allow mineral soil thermal conductivity, $M.Soil.k$, to vary with soil texture and soil moisture. We derive mineral soil conductivity following the approach of Farouki (1981) as described in Bonan (2019) (Eq. 2).

$$M.Soil.k = M.Soil.k.dry + (M.Soil.k.sat - M.Soil.k.dry) * Ke ,\qquad(2)$$

Where $M.Soil.k$ is determined by linearly ramping between saturated conductivity, $M.Soil.k.sat$ , and dry conductivity, $M.Soil.k.dry$, based on a factor, $Ke$, that varies with relative soil moisture and soil texture, represented separately for unfrozen (Eq. 3) and frozen (Eq. 4) soils.

$$Ke = \begin{cases} 1 + 0.7 * log_{10} * SE , \%Sand > 50 \\ 1 + log_{10} * SE \quad , \%Sand \le 50 \end{cases}'\qquad(3)$$

Where $Ke$ is the Kersten number, and $SE$ is the volumetric soil water content (VWC) relative to the volumetric soil water content at saturation ($VWC.sat$).

$$Ke = SE ,\qquad(4)$$

$M.Soil.k.dry$ is estimated from bulk density (Eq. 5).

$$M.Soil.k.dry = \frac{0.135pb+64.7}{2700-0.947pb} ,\qquad(5)$$

Where $pb = 2700 * (1 - VWC.sat)$. $M.Soil.k.sat$ is estimated as a function of the conductivity of solids, water, and ice in the matrix, modeled separately for unfrozen (Eq. 6) and frozen (Eq. 7) soils.

$$M.Soil.k.sat = Ksolid^{1-VWC.sat} * Kwater^{VWC.sat},\qquad(6)$$

$$M.Soil.k.sat = Ksolid^{1-VWCsat} * Kice^{VWCsat},\qquad(7)$$

We assume $Kwater = 0.57$ and $Kice = 2.29$ W m⁻¹ K⁻¹. Calculation of $Ksolid$ is calculated in Eq. 8

$$Ksolid = \frac{8.80*(\%sand)+2.92(\%clay)}{\%sand+\%clay},\qquad(8)$$

Using the total thermal resistance $R$ from Eq. 1, we can then estimate the daily sum of energy flow ($Einput;$ MJ day⁻¹) that reaches the zero isoline from the atmosphere above (Eq. 9).

$$Einput = \frac{1}{R} * (Air.Temp - Temp.zero.isoline) * \frac{86400}{1000000},\qquad(9)$$



Where $Air.Temp$ is the daily mean air temperature , $Temp.zero.isoline = 0°C$, and the constant converts from J s$^{-1}$ to MJ

day$^{-1}$. $Einput$ is then used to calculate the daily sum of water that thaws or freezes at the zero isoline based on the enthalpy (or latent heat) of fusion ($Ethaw$; 0.33 MJ/liter water). Eq. 9 is also used to estimate the daily energy flux from soil below the active layer by replacing $Air.Temp$ with the temperature of the soil below. Deep soil temperatures, here set at 5m, is assumed to be at equilibrium with mean annual air temperature of the previous decade (Riseborough, 2004).

We then model the daily amount of water that thaws or freezes, $delta.W.mm$ (Eq. 10).

$$delta.W.mm = \frac{Einput}{Ethaw},\qquad(10)$$

Where $delta.W.mm$ is constrained to values between -10 and +10 mm (only 10mm of water is allowed to freeze or thaw each day in order to avoid numerical instabilities close to the soil surface). Finally, the corresponding depth of soil that freezes or thaws each day in m, $delta.s.m$, is calculated (Eq. 11).

$$delta.s.m = \frac{delta.W.mm}{VWC.sat} * \frac{1}{1000},\qquad(11)$$

Since frozen soil (and the water captured therein) is not accessible for plants, the actual water holding capacity of the soil is dynamically modified each day. If soil thaws in a given day, that freshly melted water is added to the soil water pool and the capacity for soil to hold water increases. The approach described here also works for estimating seasonal thawing and freezing of soils in areas not underlain by permafrost.

The SOL component was adapted from Bonan and Korzuhin (1989) and Foster et al. (2019) and represents SOL depth

as a function of annual moss net primary production, biomass accumulation, respiration, and turnover. It adds live and dead moss to the fuels for forest fires, and the depth of the SOL influences post-fire tree regeneration. Annual moss productivity is simulated as a function of environmental scalars that represent effects of light attenuation through the forest canopy and moss layer and growth inhibition from fresh deciduous litter. The amount of light that reaches moss for photosynthesis attenuates with increasing forest canopy cover and with increasing moss biomass. Effects of light attenuation are represented by first

calculating the amount of light available for photosynthesis in year $t$ as $Light.avail_t$ (Eq. 12).

$$Light.avail_t = e^{-k*(LAI.forest_t+LAI.moss_t)},\qquad(12)$$

Where $k$ is the light extinction coefficient, set at 0.7, $LAI.forest_t$ is the leaf area index (m$^2$ leaf area m$^{-2}$ ground) of tree cover in year $t$. $LAI.moss_t$ is the leaf area index of moss in year $t$ calculated as moss biomass multiplied by the specific leaf area of moss (1 m$^2$ kg $^{-1}$). The effect of light attenuation on moss productivity, $FLight.avail_t$, is then calculated (Eq. 13).

$$FLight.avail_t = \frac{(Light.avail_t - LR_{min})}{(LR_{max}-LR_{min})},\qquad(13)$$

Where $LR_{max}$ is the light saturation point, or the amount of light, relative to the light level above the canopy, above which, an increase in light does not increase moss GPP; set at 0.05. $LR_{min}$ is the light compensation point, or the amount of light, relative to light level above the forest canopy, beyond which moss begins to photosynthesize; set at 0.01.





Field experiments show that fresh leaf litter from deciduous broadleaf tree species strongly inhibits moss productivity
(Jean et al., 2020). Such inhibitory effects, $FDecid_t$, are modeled as (Eq. 14).

$$FDecid_t = e^{-0.45*Decid.b_{t-1}},\qquad\qquad\qquad\qquad\qquad (14)$$

When $Decid.b_t > 0$ or 1 when $Decid.b_t = 0$. Where $Decid.b_{t-1}$ is the fresh (previous year's) forest floor deciduous litter
biomass in Mg ha$^{-1}$. $A_t$, annual assimilation by moss in year $t$ (kg biomass m$^{-2}$ leaf area) is then computed (Eq. 15).

$$A_t = A_{Max} * FLight.avail_t * FDecid_t,\qquad\qquad\qquad\qquad (15)$$

Where $A_{Max}$, the maximum moss productivity per unit leaf area, is 0.3 kg m$^{-2}$ year$^{-1}$ (Foster et al., 2019). We estimate effective
assimilation in year $t$, $A.eff_t$ in kg kg$^{-1}$ biomass. (Eq. 16).

$$A.eff_t = SLA * A_t,\qquad\qquad\qquad\qquad\qquad\qquad (16)$$

Moss productivity in year $t$, $P_t$, in kg m$^{-2}$ biomass then depends on turnover, $T_t$, and respiration, $R_t$, in year $t$ (Eq.
17-19).

$$P_t = A.eff_t * Moss.b_{t-1} - T_t - R_t,\qquad\qquad\qquad\qquad (17)$$

$$T_t = Moss.b_{t-1} * b,\qquad\qquad\qquad\qquad\qquad\qquad (18)$$

$$P_t = R_t = Moss.b_{t-1} * q,\qquad\qquad\qquad\qquad\qquad (19)$$

Where $Moss.b_{t-1}$ is the previous year's moss biomass in kg m$^{-2}$ and $b$ and $q$ are empirical parameters set at 0.136 and 0.12,
respectively (Foster et al., 2019). The moss biomass pool is updated (Eq. 20).

$$Moss.b_t = Moss.b_{t-1} + P_t,\qquad\qquad\qquad\qquad\qquad (20)$$

Note that the biomass pool can shrink if Pt becomes negative, e.g., due to a closing canopy.

Thickness of the live moss layer is calculated as biomass divided by a bulk density of 31 kg m$^{-3}$ (Walker et al., 2020).
Dead moss and forest floor litter layer thickness is calculated as biomass divided by bulk density, set at 91 kg m$^{-3}$ (Walker et
al., 2020).

The permafrost and SOL module is implemented in C++ for computational efficiency and is relatively compact (<
1,000 lines of code). It is compatible with PC or Mac and full full source code and documentation is available under a GNU
General Public License (GNU GPL www.gnu.org/licenses/gpl-3.0.html) (See code availability section).While the design is
modular, we note that the complex feedbacks between vegetation, permafrost dynamics, and SOL accumulation may require
some adaptations and code modifications when integrating our work in different forest models. Below, we detail the integration
into the individual-based forest landscape and disturbance model iLand.





**2.2 Coupling the permafrost and SOL with iLand**

iLand simulates the growth and mortality of individual trees in spatially explicit landscapes as a function of canopy light interception, climate, nutrient availability, and disturbance (Seidl et al., 2012a, b). The model was originally designed to study effects of natural disturbances, like forest fire, on forest landscapes in the context of climate change (Seidl et al., 2012a). Thus, iLand emphasizes representation of disturbances and the processes that underpin forest responses to disturbance, including tree-seed production and dispersal, abiotic filters of tree-seedling establishment, and multiple pathways of tree mortality (Seidl et al., 2012a, b; Hansen et al., 2018, 2020). For an exhaustive technical description of iLand, including carbon cycling and simulation of forest fire, see Appendix A and https://iland-model.org/, which includes full model source code.

The proportion of moss biomass that turns over (dies) each year in the new module is fed into the litter layer of iLand's decomposition module. iLand simulates decomposition as a function of climate and pool-specific carbon to nitrogen ratios (Seidl et al., 2012b). The C:N ratio of moss litter is set at 30 (Melvin et al., 2015). Together, live moss, dead moss and forest floor litter layers comprise the SOL in iLand. Wildfire ignition, spread, and severity are partially contingent on downed fuel availability in iLand (Seidl et al., 2014a), and we now include live and dead moss as available fuel in the fire module. When a grid cell burns, the combusted forest floor litter, dead moss, and live moss pools are subtracted from SOL depth.

The tree species that establish in years following fire shape multi-decadal successional trajectories (Seidl and Turner, 2022). The depth of burning in the SOL is an important determinant of seedling establishment success because the SOL is often dry and seedlings must expand their roots into mineral soil to access water (Johnstone and Chapin, 2006; Brown and Johnstone, 2012). We therefore included the effect of deep SOL as an additional limiting factor when calculating tree-seedling establishment in iLand. For each 1-ha iLand cell, the probability of establishment is scaled with a negative exponential function following Trugman et al. (2016) (Eq. 21).

$$estab.p_t = e^{-c*SOL.depth_t}, \qquad (21)$$

Where $estab.p_t$ is a multiplicative factor reducing the abiotic establishment probability in year $t$, $SOL.depth_t$ is the depth of the SOL (cm) in year $t$, and c is a species-specific shape parameter, set at 0.50 for trembling aspen (*Populus tremuloides* Michx.) and Alaskan birch (*Betula neoalaskana* Sarg.), 0.25 for white spruce (*Picea glauca* (Moench) Vass), and 0.15 for black spruce (*Picea mariana* (P. Mill.) B.S.P.).

**3 Model benchmarking**

We used a pattern-oriented modeling framework (Grimm et al., 2005) to evaluate the new module by simulating forests of interior Alaska at stand and landscape scales over days to centuries. Pattern-oriented modeling is an approach to benchmarking where patterns of many variables operating at multiple temporal and spatial scales are compared to observational datasets. We chose interior Alaska because it is located in the discontinuous permafrost zone where permafrost presence, moss production, and SOL accumulation vary with dominant forest type, disturbance history, and topography. For example, areas



dominated by mature black spruce in lowland valley bottoms and north facing slopes are generally underlain by permafrost
and support a relatively productive forest-floor moss layer and thick SOLs. Upland and south-facing slopes are dominated by
deciduous trembling aspen and Alaskan birch, which are often not underlain by permafrost, and moss is far less prevalent.
White spruce also inhabits upland positions, on its own, or mixed with black spruce, and contains SOLs of intermediate
thickness (Van Cleve and Viereck, 1981). The multiple interacting biotic and abiotic drivers of permafrost and moss
productivity create complex landscape mosaics (Johnstone et al., 2010) that we wanted to ensure the module could produce.

We first evaluated whether the module could generate realistic daily patterns of snow accumulation/melting and active
layer thawing/freezing at the stand level. We then simulated a ~61,000 ha forested landscape to test whether the approach
could realistically generate complex mosaics of near-surface permafrost presence, moss productivity, and SOL accumulation.
To ensure robust simulations, we updated an existing iLand tree-species parameter set for interior Alaska (Hansen et al., 2021)
(Table B1) and parameterized the iLand carbon cycle (Table B2) using values derived from the literature.

### 3.1 Temporal patterns of snow and active layer depth

To evaluate whether the module could generate realistic intra- and inter-annual patterns of snow accumulation and
active layer depth, we selected 17 forested sites in interior Alaska that were instrumented with temperature probes to measure
daily soil temperature at depths of zero to six m between 2014 and 2018 (https://permafrost.gi.alaska.edu/sites_list). Seven of
the sites were recorded as having an annual maximum active layer depth of less than 2 m (permafrost present). Ten of the sites
had an annual maximum active layer deeper than 2 m (permafrost absent). We used the 2 m depth cutoff because it is the
maximum effective soil depth assumed in iLand. The sites were initialized from field inventories selected to match the species
composition recorded in the soil temperature database (Walker and Johnstone, 2014; Johnstone et al., 2020). Soil information
used to initialize iLand were extracted from the global SoilGrids250m V. 1.0 (for effective soil depth) and 2.0 (for % sand,
silt, and clay) (Hengl et al., 2017). Relative soil fertility, expressed as plant available nitrogen, was set to 45 kg ha$^{-1}$ yr$^{-1}$ (Hansen
et al., 2021). Depth of the SOL was not recorded in the soil temperature database for the 17 sites. Thus, we used photos from
the instrumented sites and information on the dominant forest type to assign initial SOL depths to the iLand stands. Sites where
researchers recorded dominance of deciduous trees, or where SOLs appeared absent or shallow in photographs were assigned
a depth of 0 or 0.07 m to match independent field estimates of SOL depths in deciduous forests (Melvin et al., 2015). Sites
dominated by black spruce, or where photographs suggested a deep SOL, were assigned a depth of 0.25 m based on field
surveys of black spruce stands (Johnstone et al., 2010). Stands dominated by white spruce were assigned an intermediate depth
of 0.16 m.
Stands were simulated in iLand with 2001-2018 daily climate (minimum and maximum daily temperature,
precipitation, shortwave solar radiation, and vapor pressure deficit) from the 1-km Daymet product (Thornton et al., 2021).
We benchmarked simulated maximum annual snow depth and timing of snow melt for the period 2001-2017 (the period when
snow observations were available) using a gridded snow product (Yi et al., 2020). This product was developed by integrating
downscaled reanalysis data with satellite imagery to provide a continuous estimate of snow depth at 1-km spatial grain. When





compared with a meteorological station network (SNOTEL), the gridded observational product had a RMSE of 0.32 m with a bias of -0.09 m in mid-elevations (400-800m) where many of our forested sites were located (Yi et al., 2020).

We compared simulated and observed daily changes in active layer depth for 2014-2018, the period where soil temperature observations were available, at the seven permafrost sites and maximum annual freezing depth for the 10 non-permafrost sites with root mean squared error (RMSE). We converted observed daily soil temperatures at depths of 0.03, 0.5,

1, 1.5, 2, 4, and 6 m to active layer depth by identifying the zero isoline with linear interpolation. We also compared the day of year when maximum active layer depth and freezing depth were reached in simulations and observations.

### 3.2 Landscape heterogeneity in near-surface permafrost presence, moss productivity, and SOL accumulation

We evaluated whether the module, coupled with iLand, could simulate landscape-scale mosaics of near-surface permafrost (≤ 1 m deep), moss production, and SOL accumulation in a large forested area (~61,000 ha of land area). We

initialized the model with a tree-species composition map based on a remotely sensed plant functional type (PFT) product that classified vegetation as spruce, deciduous, mixed forest, or non-forest (Wang et al., 2020) and reflected fire history. We further decomposed PFTs into black spruce, white spruce, trembling aspen, Alaskan birch, mixed forest, potential forest (i.e., areas currently unforested that could support forest in the future), and nonforest using rules based on aspect, elevation, and a permafrost map (Table B3). While this approach allowed us to disaggregate PFTs to the species level, we lack robust datasets

to evaluate the accuracy of the species composition map. This is a challenge as dominant tree species determines SOL accumulation and permafrost distribution. In the future, well validated remotely sensed tree-species composition maps would markedly reduce initial condition uncertainty of forest simulations in interior Alaska (Hermosilla et al., 2022).

Initial stand densities, tree sizes, and forest-floor carbon pools (litter, coarse wood, live and dead moss; Table B4) for the appropriate tree species were initialized in the model as early postfire (11 years old) forest based on field inventories

(Walker and Johnstone, 2014; Johnstone et al., 2020). Because the forest landscape was initialized as entirely early postfire, it did not reflect variation in forest stand age. Thus, we ran a 200-year spin up as a function of historical climate (climate years 1950-2005 recycled randomly with replacement) and simulated fire dynamically to generate spatial heterogeneity consistent with internal model logic, following protocols established in previous iLand studies (Hansen et al., 2020; Turner et al., 2022). We then simulated forests for another 100 years and used this period in all analyses.

We want to eventually conduct simulations with future 21st century climate. Thus, we used daily meteorological data from the historical period of the CMIP5 generation CCSM4 General Circulation Model (GCM) (Gent et al., 2011) to force landscape-level simulations instead of DAYMET (as was used in the stand-level experiment). This GCM corresponds closely with observed historical climate in Alaska (Walsh et al., 2018), and we statistically downscaled it to a 1-km spatial resolution using quantile matching with Daymet as the observational grid (Hansen et al., 2021).We extracted soils data from the same

sources as the stand-level experiment that geographically corresponded to the 1-ha grid-cells in our simulated landscape. Because fire is stochastic in iLand, and an important determinant of permafrost dynamics, SOL depth, tree-species





composition, and stand structure, we ran ten replicates and analyzed output from the run with the smallest difference between modeled and observed mean annual burned patch size and annual probability of a fire event.

We compared fire from simulation years 201-300 to observations in the Alaska Large Fire Database from the period
1980 - 2021. This database contains perimeters for larger fires (size threshold for inclusion has varied over time, ranging from 10-1,000 ha) and point locations for smaller fires. We combined these datasets to ensure comprehensive coverage and assumed a circular shape for the smaller fires when perimeters were unavailable. Fire is a stochastic process in iLand, so, we did not expect perfect correspondence between modeled and observed individual fire sizes and locations. Instead, we aimed for the model to generate fire characteristics (i.e., frequency, patch size, annual area burned, and severity) that were generally
consistent with the observational record. We took two approaches for benchmarking. First, we compared simulated and observed annual probability of fire occurrence and mean annual burned patch size, as well as the proportion of stems and basal area killed by fire. Second, we compared simulated and observed fire characteristics from the landscape with observed fire characteristics in all of forests of interior Alaska broken into 625 ~ 61,000ha landscapes. This allowed us to determine how the dynamic fire module in iLand performed for our landscape, specifically, and how the model performed relative to the
spatial variation in fire regimes across interior Alaska.

We compared the proportion of the landscape underlain by near-surface permafrost in the last 30 years of simulation (years 271-300) to a remotely sensed product of near-surface permafrost presence (Pastick et al., 2015). This product was created by integrating satellite records and other geospatial datasets to predict the probability of near-surface permafrost presence at a 30m spatial resolution with machine learning. Because iLand operates at 1-ha spatial resolution for permafrost,
we aggregated the remotely sensed data from 30-m to 1-ha grid cells by calculating the mean probability of near-surface permafrost presence in each 1-ha grid cell. We then used a $\geq 50\%$ probability of permafrost presence, the same cutoff used in the original analysis (Pastick et al., 2015), to map the permafrost distribution. In iLand, near-surface permafrost was considered present in any grid cell where the annual maximum active layer depth was $\leq 1$ m in 50% of years in the last 30 years of simulation. This cutoff ensured we only included areas that were underlain by frozen ground most years. We compared the
total proportion of the landscape underlain by near-surface permafrost and how permafrost presence varied as a function of aspect in simulations and the benchmarking product. We also evaluated how permafrost presence varied as a function of simulated dominant tree species, but did not compare to the benchmarking product because we lack tree species composition maps in interior Alaska.

We compared SOL carbon in simulation year 300 separated by forest type to field inventories (Alexander and Mack,
2016; Walker et al., 2020). While benchmarking data was unavailable, we also evaluated landscape variability in total SOL and live moss depth. We assessed SOL combustion by fire in different forest types for model years 260-300, to ensure a sufficient number of fires, as compared to two extensive sets of postfire field plots (Walker and Johnstone, 2014; Johnstone et al., 2020; Walker et al., 2020).





Because near-surface permafrost presence and moss productivity are affected by and feedback to influence forest
dynamics, we determined whether the model could realistically represent landscape-level patterns of tree-species composition
and stand structure.

We explored how landscape patterns of dominant forest type shifted through 300 years of simulation and compared
simulated stand density and basal area of each forest type from the end of the simulation with two field inventories. The first
was a regional network of permanent plots in interior Alaska collected by the Bonanza Creek Long Term Ecological Research
Network site (Ruess et al., 2021). The second inventory was the Cooperative Alaska Forest Inventory, which is a set of
permanent plots covering interior Alaska, south-central Alaska, and the Kenai Peninsula (Malone et al., 2009).

We also compared simulated aboveground live tree biomass from the end of the simulation with remotely-sensed
estimates of aboveground live woody biomass for the same landscape (Wang et al., 2021). This dataset is a 30-m product that
characterizes annual live woody biomass for the years 1984-2014. We aggregated 2014 biomass estimates to the 1-ha spatial
resolution of iLand using bilinear interpolation. We further benchmarked snag and coarse wood carbon pools in model year
300 with published field observations (Alexander and Mack 2015, Melvin et al. 2015).

To quantify the underpinning drivers of landscape variability in tree-species composition and aboveground live and
dead biomass, we compared simulated variation in postfire tree-seedling density from years 260-300 by species and SOL depth
with field observations (Walker and Johnstone, 2014; Johnstone et al., 2020). Finally, we analyzed the computational
325   efficiency of the module by simulating the landscape with and without the permafrost module turned on to quantify its memory
requirement and run time.

Dominant forest type was determined using species importance values (IV), a measure of stand dominance based on
the relative proportions of species density and basal area. It ranges from zero to two (Hansen et al., 2020). We considered
stands dominated by a particular species if their IV was greater than one. Stands were considered mixed-spruce or mixed-
330   deciduous forest if black sprue and white spruce or aspen and birch IVs summed to greater than one, respectively. Averages
in the text are presented as medians and inter-quartile ranges (IQRs) (25th-75th percentiles). Benchmarking analyses were
conducted in R statistical software V. 4.0.4 (R Core Team, 2021) using the packages tidyverse (Wickham et al., 2019) and
terra (Hijmans, 2021).

## 4 Results

335   **4.1 Snow depth, timing of snow melt, and active layer depth**

Simulated maximum annual snow depth corresponded with observations when the module was forced with 2001-
2017 climate, but overestimated snow depth for sites and years where snow fall was above average (RMSE of 0.33m) (Fig.
2A). The model closely captured timing of spring snow melt (RMSE of 8.5 days) (Fig. 2B).

When forced with 2014-2018 climate, simulated daily patterns of active layer depth (Fig. 3) and maximum annual
active-layer depths closely matched observations from seven forest stands underlain by permafrost (RMSE of 0.37 m) (Fig.





2C). On average, maximum annual active-layer depth occurred 15 days later in iLand than in observations. The model also reasonably recreated maximum annual freezing depths at 10 forest stands not underlain by permafrost (RMSE of 0.44 m) (Fig. 2D). On average, the maximum annual freeze depth was reached 11.3 days earlier in simulations than in observations.

### 4.2 Landscape-level fire characteristics

Mean annual burned patch size was 3,628 ha and annual probability of a fire event was 11% in the best of the ten replicate landscape simulations and differed from observed values by only 5% and 8%, respectively (Fig. 4A, 4B). However, among all ten replicates, burned patch size and probability of a fire event differed by as much as 44% and 42%, highlighting the stochastic nature of fire. Both observed and simulated fire metrics for the landscape were also representative of observed fire characteristics in all 625 sampled 61,000 ha landscapes across the boreal domain of Alaska (Fig. 4C). On average, 73 (70

– 76) % of stems and 52 (46 – 60) % of basal area was killed by fire in the model (Fig. B1).

### 4.3 Landscape near-surface permafrost, moss, and SOL depth

The model simulated 34.6% of forested area in the landscape as underlain by permafrost between years 271-300; nearly identical to the estimate of 33.4% of forested area from the benchmarking product (Fig. 5A). Aspect was an important determinant of permafrost presence in the model and in observations (Fig. 5B). Near-surface permafrost presence also varied

with dominant tree species in iLand. Seventy-three percent of simulated black spruce forest area was underlain by near surface permafrost, followed by 11% of mixed spruce, 9% of white spruce forest, 2% of aspen dominated stands, 1% of mixed deciduous forest, and 0.1% of birch dominated forest.

Soil-surface organic layer C in simulation year 300 averaged 4,801 (2,965 – 6,575) g m$^{-2}$. When broken out by dominant forest type, simulated SOL C closely corresponded to observations for all forest types where comparison was

possible (Fig. 6A). Dead moss and litter depth across the landscape averaged 11.6 (7.4 - 15.5) cm in simulation year 300, and live moss depth averaged 5.4 (2.7 – 8.7) cm, with pronounced spatial heterogeneity (Fig. 6B). Tree species composition was an important determinant of total SOL depth (Fig. B2): The SOL was thickest in black spruce-dominated stands, averaging 25 (21 – 27) cm, followed by white spruce; 17 (15 – 19) cm, mixed spruce; 14 (7 – 17) cm, aspen; 9 (4 – 11) cm, mixed deciduous; 5 (4 – 10) cm, and birch dominated forest; 4 (3.7 – 4.1) cm. Fire occurrence also strongly influenced SOL depth. In black and

white spruce stands, fire combusted 9 (6 – 12) cm on average. In contrast, almost no SOL was combusted in deciduous stands.

### 4.4 Landscape-level tree species composition and forest structure

Between simulation year 0 and 300, forest cover increased from 48,811 ha to 60,629 ha, as trees colonized areas initialized as potential forest. The model was initialized with black spruce forest comprising 41% of the land area, followed by white spruce (22%), aspen (7%), birch (6%), and mixed forest (5%) (Fig. 7A). By year 300, the land area dominated by

black spruce remained high at 40% (Fig 7B). However, white spruce-dominated forest area declined markedly to 2% because black spruce trees colonized white-spruce stands, as is commonly found in interior Alaska (Van Cleve and Viereck, 1981;





Burns and Honkala, 1990). At the end of the simulation, mixed spruce stands comprised 42 % of land area. Aspen and birch also intermixed by year 300, with mixed-deciduous forest covering 11% of the landscape. The land area dominated by aspen in year 300 declined to 2%, and birch-dominated forest declined to 3% of the landscape.

Stand density and basal area in the model corresponded well with multiple field observation datasets in year 300 (Fig. 8). Aspen and birch stands were most dense, followed by black spruce and white spruce dominated stands. Deciduous dominated stands also had the greatest basal area, followed by white spruce and black spruce stands (Fig. 8B). Simulated aboveground live woody biomass across the landscape was within 28% of the observed average. Aboveground live woody biomass in iLand was 51,931 (20,456 – 68,200) kg ha$^{-1}$, on average, and observed biomass was 39,277 (9,219 – 56,246)

kg ha$^{-1}$. Average simulated standing snag carbon differed from the observed average by 41% (Fig. B3A). Simulated downed coarse wood C varied markedly by dominant forest type and corresponded closely to field observations (Fig. B3B).

Simulated tree-seedling density two years after fires closely matched field observations and varied with depth of postfire SOL (Fig. 9). Birch and aspen seedlings were most abundant where SOLs were shallow (0-5cm), with 8.9 (6.1 – 13.1) and 6.2 (4.9 – 8.0) seedlings m$^{-2}$ establishing. Black spruce seedlings were the next most abundant at 3.6 (0.3 – 4.8) seedlings

m$^{-2}$, followed by white spruce with 0.3 (0.2 – 0.4) seedlings m$^{-2}$. Where SOLs were thicker (15-20 cm), black spruce density averaged 2.1 (1.3 – 3.1) seedlings m$^{-2}$, and aspen, white spruce, and birch rarely established.

**4.5 Computation efficiency**

The memory footprint of the permafrost and SOL module was approximately 15 MB (~ 0.1% of total memory footprint), and it increased overall run time by 1%.

**5 Conclusions**

Ecological legacies will determine how forests are affected by climate change and increasingly prevalent disturbances, like fire (Turetsky et al., 2016; Kannenberg et al., 2020; Hansen et al., 2022). However, some legacies uniquely important to the structure and functioning of boreal forests (e.g., permafrost and SOLs) are rarely considered in models used to project 21$^{st}$-century ecological change. Here, we present a new permafrost and SOL module that operates at fine temporal

(daily) and spatial (1-ha) scales and is computationally efficient. The module simulates daily changes in active layer depth, moss production, and annual SOL accumulation (Fig. 1). When coupled to a forest model, it also represents the complex ecological effects of permafrost and SOLs on boreal forests and fire. Benchmarking results demonstrate the model recreates temporal and spatial patterns consistent with observations at stand to landscape scales over days to centuries. Our model will contribute to improving 21$^{st}$-century projections of boreal forest change.

Process-based simulation models are powerful tools for assessing how forests will change (Seidl, 2017; Albrich et al., 2020; Fisher and Koven, 2020). Forests often respond slowly to stressors relative to other ecological systems (Hughes et al., 2013; Turner et al., 2022). As a result, models must capture dynamic feedbacks among variables and represent the key



legacies that accumulate over decades to centuries in order to project future trajectories of forests (Johnstone et al., 2016). Our objective was to mechanistically represent permafrost and SOLs and capture effects of daily variability in weather as well as

the feedbacks that arise among forest dynamics, fires, and permafrost in topographically complex landscapes. The model was skilled at capturing daily patterns of freezing and thawing as well as the inter-annual variability in maximum thaw depth and the timing of maximum thaw. The model also reasonably recreated snow accumulation patterns for most years, but overestimated depth in years where snow pack was unusually deep, likely because we used a single snow-density parameter value. In reality, snow density varies tremendously across landscapes and over time. iLand takes a relatively simple approach

to simulating snow derived from Running and Coughlan (1988). In the future, a more advanced snow model could be added that includes key processes affecting snow depth and conductive properties, including the representation of variation in snow density, freeze thaw cycles, and sublimation (Bormann et al., 2013; Jafarov et al., 2014).

The module was designed to represent permafrost and SOL effects on forest dynamics and fire. In particular, it determines the water available to plants and accumulation of forest floor biomass, which serves as fuels for fire and influences

postfire tree regeneration. When coupled with iLand, the model reproduced common secondary successional trajectories found in interior Alaska, including self-replacement and disturbance-induced abrupt transitions in forest types (Johnstone et al., 2010, 2016). For example, when thick SOLs remained after fire in black-spruce stands, self-replacement was common, leading to recovery of forests functionally and structurally similar to the prefire stands (Anderson et al., 2003; Johnstone and Kasischke, 2005). In contrast, when fires combusted most of the SOL in black spruce stands, abrupt transitions from spruce- to deciduous-

dominated forest (mixtures of aspen and birch) occurred, consistent with regional trends documented in the last few decades (Johnstone et al., 2010, 2020).

The boreal forest biome is warming at least two times faster than the global average (IPCC, 2021), causing climate sensitive disturbances, like fire, to increase in frequency and severity (Seidl et al., 2020; Walker et al., 2020). Our permafrost and SOL module will help process-based modelers produce more accurate projections of how forests in the biome are likely

to change over the next century. Better projections will resolve a number of important uncertainties, including 1) where increased burning due to climate change may reduce boreal fuel loads such that fire-self limitation emerges (Héon et al., 2014; Buma et al., 2022); 2) when shifts in postfire successional trajectories will initiate biophysical feedbacks that further alter regional climate; and 3) how climate change, fire, and permafrost thaw will interact to reshape boreal carbon cycling (Schurr et al., 2018; Schuur and Mack, 2018; Mack et al., 2021). Because boreal forests have disproportionate impacts on the climate

system through biogeochemical and biophysical pathways, such information is essential to inform innovative and effective global climate mitigation and adaptation strategies.





**Appendix A**

*Carbon cycling in iLand*

iLand dynamically models carbon in live foliage, branch, stem, and root compartments, and in standing snag, forest-floor litter, downed coarse wood, and mineral soil organic material pools (Seidl et al., 2012b). Primary production is simulated with a radiation use efficiency approach. Carbon fixed by trees is then allocated to different tree compartments based on allometric equations, representing functional balance. Influxes of carbon from live compartments to dead organic matter pools

are calculated based on leaf turnover rates, tree mortality, and snag dynamics. Snag fall occurs over time based on a species-specific half-life. When snags fall, they are added to the downed coarse wood pool. Decomposition of dead organic matter pools is represented with a pool- and species-specific optimal decomposition rate (10°C, no water limitation) that is then modified by prevailing temperature and precipitation.

*Forest fire in iLand*

The model also includes robust representations of several natural disturbances, including forest fire (Seidl et al., 2014a, b; Hansen et al., 2020). Fire occurrence and spread are dynamically simulated at a 20-m resolution as a function of 20th-century fire probability and size distributions, landscape topography, model-generated wind speed and direction, and the proportion of total downed litter and coarse wood pools that are burnable, which is determined by fuel moisture (as quantified by the Keetch Byram Drought Index; KBDI). For every 20-m grid cell that burns, the available fuels are assumed combusted.

Percent crown kill of live trees is estimated as a function of tree size, available fuel loads and aridity. For the portion of live tree canopies that are killed, we assumed 90% of foliage, 50% of branch, and 30% of the burned stem biomass is combusted. Tree mortality from fire is simulated probabilistically based on tree size, percent crown kill, and bark thickness; a model parameter that varies by tree species. If a tree dies, the non-combusted foliage and branches are added to the downed litter and coarse wood pools. Portions of killed tree stems that were not combusted enter the standing snag pool.







## Appendix B

**Table B1 Species parameters for interior Alaskan boreal forest. Pima=** *Picea mariana* **(black spruce), Pigl=***Picea glauca*
**(white spruce), Potr=** *Populus tremuloides* **(trembling aspen), Bene=** *Betula neoalaskana* **(Alaskan birch). dim =**
**dimensionless, exp = expression, sdlings = seedlings. See Hansen et al. 2021 for sources.**

| Parameter | Unit | Pima | Pigl | Potr | Bene |
|---|---|---|---|---|---|
| ***Tree growth*** | | | | | |
| Specific leaf area | $m^2\,kg^{-1}$ | 2.77 | 3.97 | 17 | 18.5 |
| Leaf turnover | $yr^{-1}$ | 0.05 | 0.2 | 1 | 1 |
| Root turnover | $yr^{-1}$ | 0.33 | 0.33 | 0.33 | 0.33 |
| Height to diameter low a | dim | 35.1 | 55.05 | 48.58 | 55.37 |
| Height to diameter low b | dim | -0.13 | -0.2 | -0.13 | -0.2 |
| Height to diameter high a | dim | 330.94 | 357.5 | 402.66 | 577.5 |
| Height to diameter high b | dim | -0.39 | -0.37 | -0.36 | -0.5 |
| Wood density | $kg\,m^{-3}$ | 380 | 330 | 350 | 480 |
| Form factor | dim[0,1] | 0.36 | 0.4 | 0.41 | 0.4 |
| ***Biomass allocation*** | | | | | |
| Stem wood biomass a | * | 0.1179 | 0.04844 | 0.06401 | 0.14796 |
| Stem wood biomass b | * | 1.99 | 2.51 | 2.51 | 2.25 |
| Stem foliage biomass a | * | 0.0554 | 0.02522 | 0.012 | 0.012 |
| Stem foliage biomass b | * | 1.45 | 2.04 | 1.45 | 1.45 |
| Root biomass a | * | 0.027774 | 0.02774 | 0.052813 | 0.02533 |
| Root biomass b | * | 2.289 | 2.289 | 2.204 | 2.417 |
| Branch biomass a | * | 0.0738 | 0.001194 | 0.00008 | 0.01187 |
| Branch biomass b | * | 1.3827 | 3.04738 | 4.13 | 2.4 |
| ***Mortality*** | | | | | |
| Probability of survival to max age (intrinsic mortality) | dim[0,1] | 0.1 | 0.1 | 0.1 | 0.01 |
| Stress-related mortality | dim | 1 | 1 | 1 | 1 |
| ***Aging*** | | | | | |
| Max age | years | 250 | 550 | 250 | 225 |
| Max height | m | 15 | 55 | 35 | 30 |
| Aging a | dim | 0.5 | 0.5 | 0.5 | 0.5 |



| | | | | | |
|---|---|---|---|---|---|
| Aging b | dim | 2.5 | 2.5 | 2.5 | 2.5 |
| ***Environmental responses*** | | | | | |
| Vapor pressure deficit response | dim | -0.65 | -0.65 | -0.65 | -0.5 |
| Min temperature | °C | -8 | -4 | -5 | -6 |
| Optimum temperature | °C | 13 | 20 | 17 | 15 |
| Nitrogen class | dim[1,3] | 2 | 1 | 1 | 1 |
| Phenology | int[0,2] | 0 | 0 | 1 | 1 |
| Max canopy conductance | m s⁻¹ | 0.0212 | 0.0212 | 0.0207 | 0.0207 |
| Min soil water potential | MPa | -1.5 | -3 | -3.5 | -2 |
| Light response | dim[1,5] | 4.5 | 3 | 1 | 1 |
| Fine root to foliage ratio | dim[0,1] | 0.75 | 0.75 | 0.75 | 0.75 |
| ***Seed production and dispersal*** | | | | | |
| Cone bearing age | years | 15 | 30 | 50 | 90 |
| Seed year interval | years | 1 | 2 | 4.5 | 4 |
| Non-seed year fraction | dim[0,1] | 0 | 0.003 | 0.02 | 0.02 |
| Seed mass | mg | 0.89 | 2.2 | 0.17 | 0.34 |
| Germination rate | dim[0,1] | 0.4725 | 0.029 | 0.0475* | 0.038* |
| Fecundity | sdlings m⁻² | 500 | 15 | 185* | 85* |
| Seed kernel a | m | 7 | 110 | 170 | 170 |
| Seed kernel b | m | 200 | 600 | 400 | 400 |
| Seed kernel c | dim[0,1] | 0.05 | 0.5 | 0.62 | 0.62 |
| ***Establishment*** | | | | | |
| Min temperature | °C | -69 | -70 | -80 | -80 |
| Chill requirement | days | 20 | 42 | 40 | 44 |
| Min growing degree days | degree days | 100 | 130 | 227 | 227 |
| Max growing degree days | degree days | 3,060 | 3,459 | 4,414 | 4,122 |
| Growing degree days base temperature | °C | 3.0 | 2.7 | 3.5 | 3.7 |
| Growing degree days before bud burst | degree days | 123 | 147 | 189 | 231 |
| Frost free days | days | 60 | 60 | 81 | 80 |
| Frost tolerance | dim[0,1] | 0.9 | 0.9 | 0.9 | 0.9 |
| ***Sapling growth*** | | | | | |





| | | | | | |
|---|---|---|---|---|---|
| Sapling growth a | dim | 0.03 | 0.035 | 0.12 | 0.12 |
| Sapling growth b | m | 20 | 35 | 25 | 25 |
| Max stress years | years | 5 | 2 | 2 | 5 |
| Stress threshold | dim[0,1] | 0.05 | 0.22 | 0.2* | 0.25* |
| Height to diameter ratio | dim | 88 | 85 | 170 | 119 |
| Reineke's R | saplings ha$^{-1}$ | 400 | 75 | 250 | 650 |
| Reference ratio | dim[0,1] | 0.5 | 0.637 | 0.8 | 0.55 |
| ***Serotiny*** | | | | | |
| Serotiny formula | exp | 30,0.99,80,0.99 | NA | NA | NA |
| Serotiny fecundity | dim | 30 | NA | NA | NA |
| ***Crown parameters for light influence patterns*** | | | | | |
| Crown shape coefficient | dim | 0.2593 | 0.28357 | 0.32326 | 0.33303 |
| Max crown radius a | m | 1.0302 | 1.23219 | 1.56269 | 1.64401 |
| Max crown radius b | m | 2.4095 | 3.141 | 4.338 | 4.6325 |
| Relative crown height | dim[0,1] | 0.5645 | 0.605 | 0.3815 | 0.5555 |

* Adjusted from Hansen et al. 2021 with addition of permafrost module



**Table B2. iLand carbon cycle parameters. Pima=** *Picea mariana* **(black spruce), Pigl=***Picea glauca* **(white spruce), Potr=** *Populus tremuloides* **(trembling aspen), Bene=** *Betula neoalaskana* **(Alaskan birch)**

| Parameter | Pima | Pigl | Bepa | Potr |
|---|---|---|---|---|
| Litter C:N ratio* | 73 | 73 | 17.9 | 21 |
| Fine root C:N ratio* | 45 | 45 | 45 | 45 |
| Wood C:N ratio* | 425.6 | 425.6 | 336.6 | 405.5 |
| Standing snag decomposition under optimal climate# | 0.2 | 0.2 | 0.2 | 0.2 |
| Snag half life# | 25 | 15 | 15 | 15 |
| Litter decomposition under optimal climate# | 0.23 | 0.33 | 0.39 | 0.56 |
| Coarse wood decomposition under optimal climate# | 0.06 | 0.02 | 0.15 | 0.15 |

*Alexander and Mack 2015; # This study



**Table B3. Rules for converting plant functional type maps from Wang et al. (2020) to species level maps for initializing iLand.**

| Species | Rule |
|---|---|
| Black spruce | • PFT is spruce and aspect is north<br>• PFT is spruce, aspect is flat, and permafrost is present<br>• PFT is woodland |
| White spruce | • PFT is spruce and aspect is *not* north<br>• PFT is spruce, aspect is flat, and permafrost is *not* present |
| Trembling aspen | • PFT is deciduous and aspect is south |
| Alaskan birch | • PFT is deciduous and aspect is not south |
| Mixed forest | • PFT is mixed forest |
| Potential forest | • PFT is low shrub, tall shrub, open shrub, herbaceous, or tussok tundra |





**Table B4. Initial conditions for iLand carbon cycle. Pima=** *Picea mariana* **(black spruce), Pigl=***Picea glauca* **(white spruce), Potr=** *Populus tremuloides* **(trembling aspen), Bene=** *Betula neoalaskana* **(Alaskan birch).**

| State variable | Unit | Pima | Pigl | Bepa & Potr | Mixed forest | Sources |
|---|---|---|---|---|---|---|
| Forest floor moss | Kg biomass ha⁻¹ | 25,000-45,000 | 5,000-15,000 | 10 | 10,000 - 25,000 | Johnstone et al. 2020 / Walker et al. 2014 |
| Forest floor leaf litter, dead moss, and fine roots | Kg C ha⁻¹ | 48,682-96,901 | 48,682-96,901 | 17,371-31,765 | 33,026.5-64,336 | Alexander and Mack 2015 |
| Coarse downed wood coarse root C | Kg C ha⁻¹ | 17,000 | 17,000 | 20,020 | 18,500 | Alexander and Mack 2015 |
| Organic C in mineral soil | Kg C ha⁻¹ | 35,000 | 35,000 | 35,000 | 35,000 | Melvin et al. 2015 |





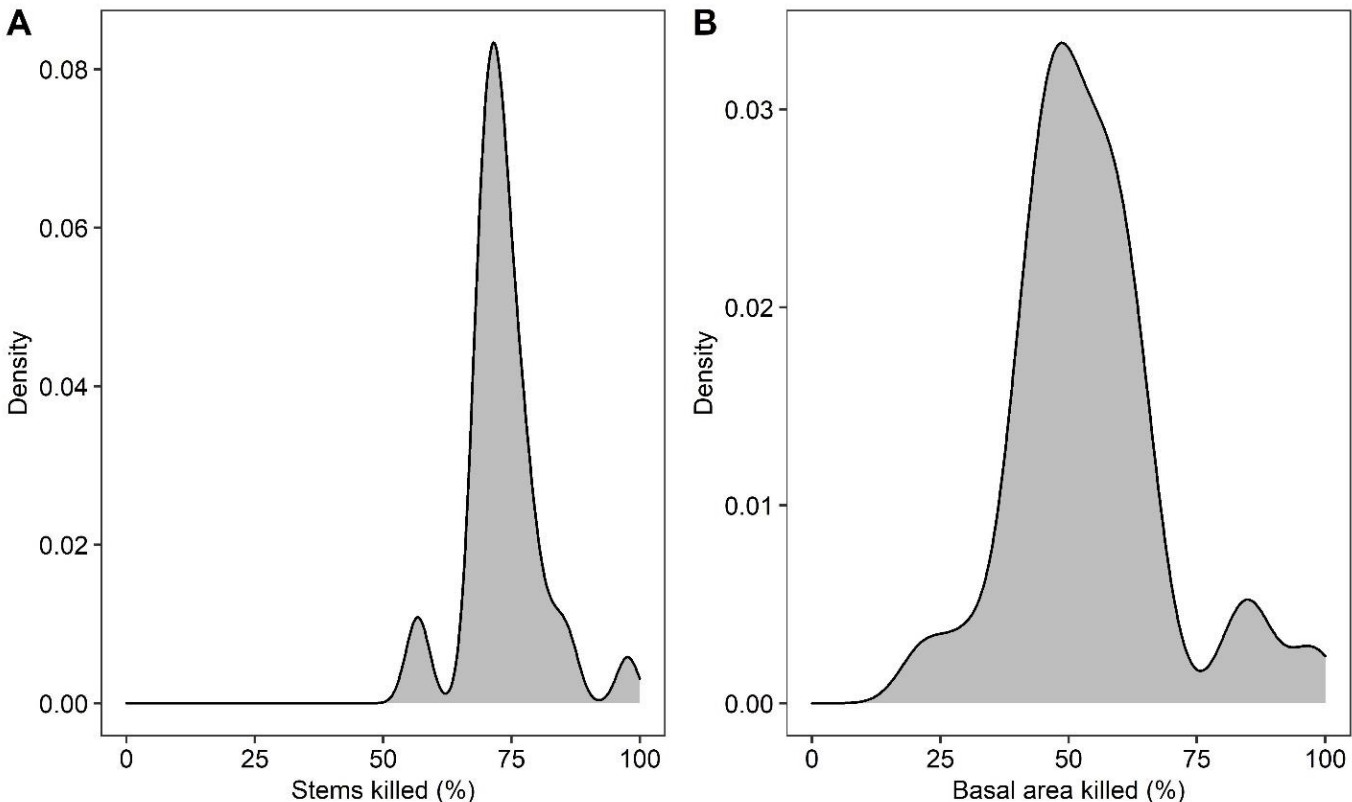

**Figure B1. Simulated percent of A. stems killed and B. basal area killed by fire in a 61,000 ha forested landscape in**
**interior Alaska. Model output is from simulation years 201- 300.**



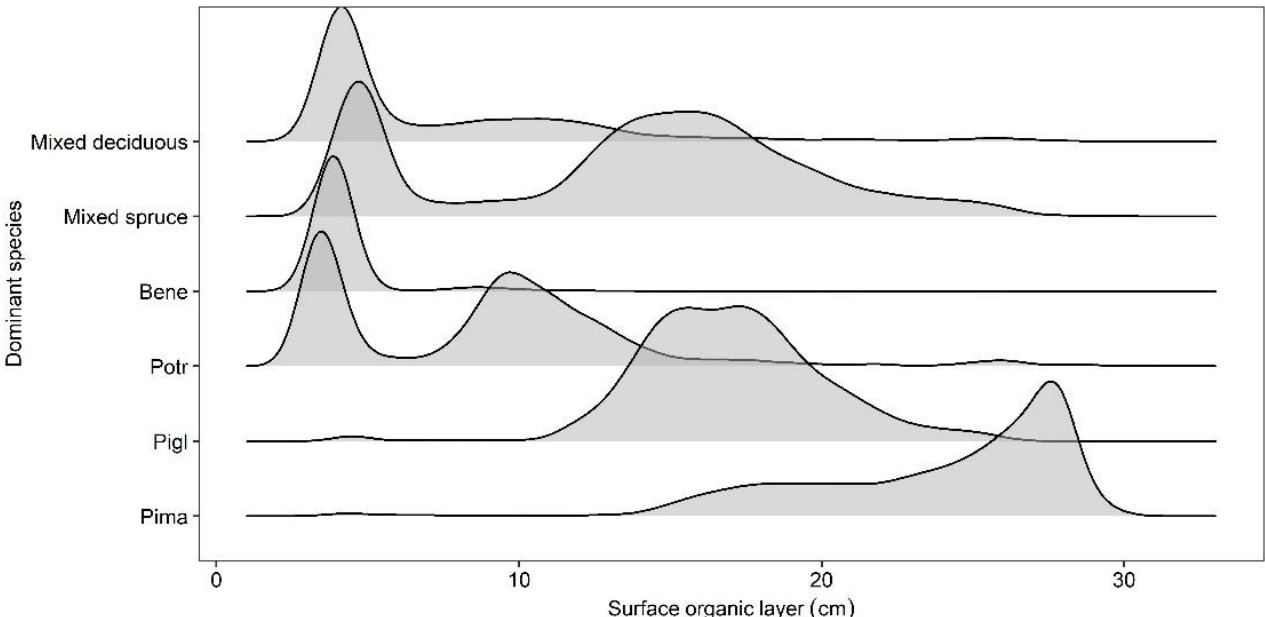

**Figure B2. Simulated surface organic layer depth as a function of dominant forest type in a 61,000 ha forested landscape**
**in interior Alaska. Model output is from simulation year 300.**





**Figure B3. Observed and simulated A. standing snag carbon and B. downed coarse wood carbon as a function of dominant forest type. Bars and whiskers show means ± 1 standard deviation in plot A and means ± 1 standard error**

**due to the limited availability of the raw field observations. Modeled carbon stocks are from simulation year 300 in a 61,000 ha forested landscape in interior Alaska. Observations are from field sampling in other boreal forest stands.**



**Code and data availability**

The source code is available as a supplement to this paper. The model executable and source code, project directories, and
analysis R scripts used in this project are also available at the Cary Institute of Ecosystem Studies data repository (DOI:
https://doi.org/10.25390/caryinstitute.21339090). A technical description of the permafrost and SOL module is available at
https://iland-model.org/permafrost.

**Author contributions**

WR and WDH developed the permafrost and SOL module, WDH conducted benchmarking simulations, analyzed outputs, and
wrote the paper. All co-authors contributed to the paper.

**Competing interests**

The authors declare that they have no conflict of interest.

**Acknowledgements**

We are grateful to Brendan Rogers, Scott Goetz, Michelle Mack, and Xanthe Walker who provided feedback on an earlier
draft of this paper. WDH acknowledges support from the National Science Foundation (Grant # OPP 2116863) and the Royal
Bank of Canada. RS and WR acknowledge funding from the European Research Council under the European Union's
Horizon 2020 research and innovation program (Grant Agreement 101001905). BVG acknowledges support from the Joint
Fire Sciences Program (Project: 20-2-01-13).

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





**Figure 1. Conceptual diagram of the permafrost and soil-surface organic layer module. State variables are in white, processes are described in black, and forcing variables are in red.**







**Figure 2. A. Relationship between observed and simulated maximum annual snow depth at 20 sites between 2001-2017**

**B. Relationship between observed and simulated day of spring snow melt at 20 sites between 2001-2017. C. Relationship between observed maximum annual thaw depth and simulated maximum annual thaw depth at 7 sites underlain by permafrost between 2014-2018 (only site-years with complete observational records are included). D. Relationships between observed and simulated maximum annual freeze depth at 10 sites not underlain by permafrost between 2014-2018 (only site-years with complete observational records are included).**




**Figure 3. A. Daily active layer freezing and thawing in 2016 at one of seven forest stands underlain by permafrost. B Daily thawing and freezing in 2016 at one of ten forest stands not underlain by permafrost. Solid lines represent snow depth. Dots represent active layer depth or depth of freezing. Grey fill represents frozen soils. Blue fill represents unfrozen soils**





**Figure 4. Simulated and observed A. annual fire probability and B. mean burned patch size in a 61,000 ha landscape in interior Alaska. Model output is for years 201-300. Observations are from years 1980-2020. The grey density distribution shows observed values for all sampled 625 61,000 ha landscapes across the boreal domain of interior**

**Alaska. C. Map of all 625 sampled landscapes. Red dot shows the landscape simulated in iLand.**





**Figure 5. A. Observed and simulated near-surface (≤ 1m deep) permafrost in a 61,000 ha forested landscape in interior Alaska. B. Observed and simulated percent of forested area underlain by near-surface permafrost in the same landscape as a function of aspect. Simulated permafrost presence is for years 271-300 of the simulation.**







**Figure 6. A. Observed and simulated surface organic layer carbon as a function of dominant forest type. Bars and whiskers show mean SOL carbon ± 1 standard error due to limited availability of raw observational data. Simulated**

**SOL carbon is from simulation year 300 in a 61,000 ha forested landscape in interior Alaska. Observations are from field sampling in other boreal forest stands. B. Simulated dead moss and tree litter depth as well as live moss depth are from simulation year 300 in a 61,000 ha forested landscape in interior Alaska. Together, these two variables comprise the total surface organic layer in iLand.**



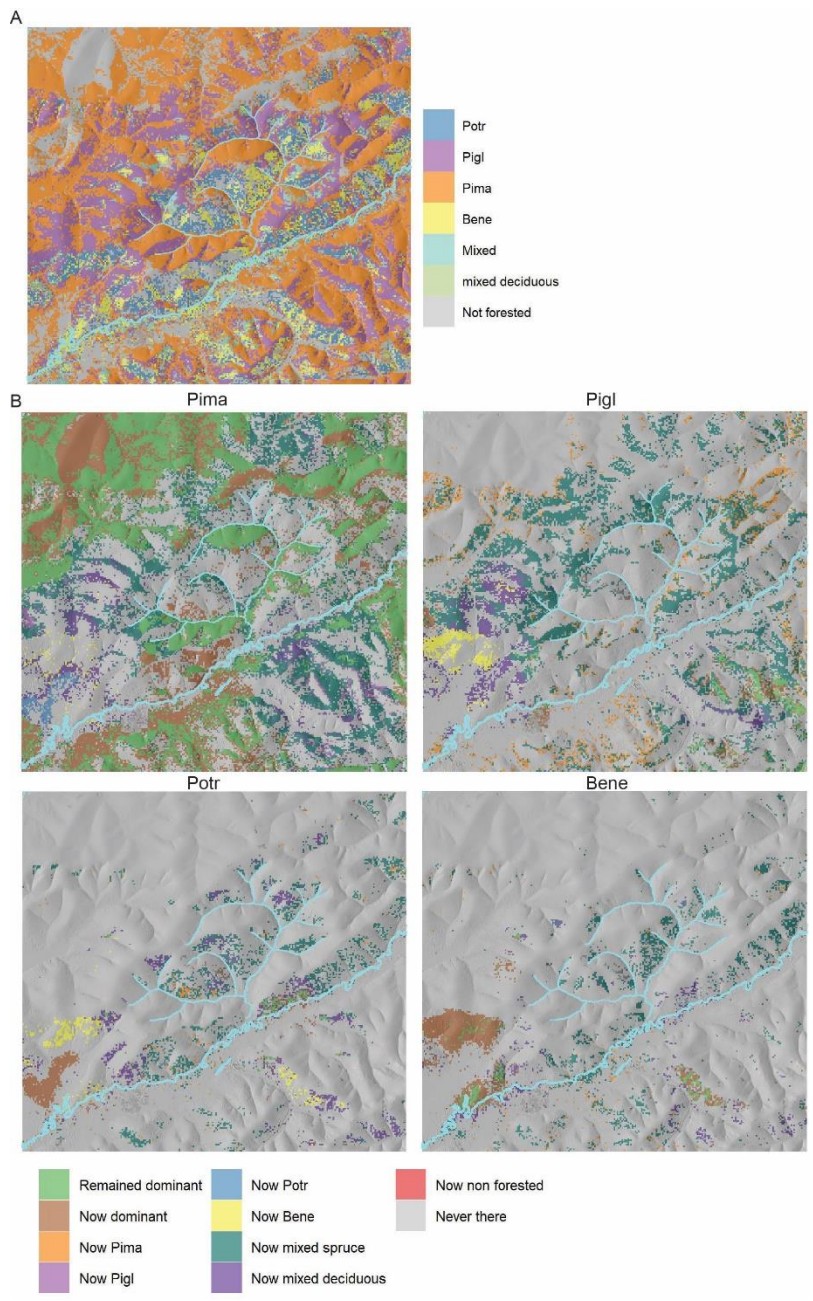


**Figure 7. A. Tree-species composition in a 61,000 ha forested landscape of interior Alaska used to initialize iLand. B. Maps showing how forested area initially dominated by each tree species changed over 300 years of simulation. Pima (*Picea mariana*) = black spruce, Pigl (*Picea glauca*) = white spruce, Potr (*Populus tremuloides*) = trembling aspen, Bene (*Betula neoalaskana*) = Alaskan birch.**



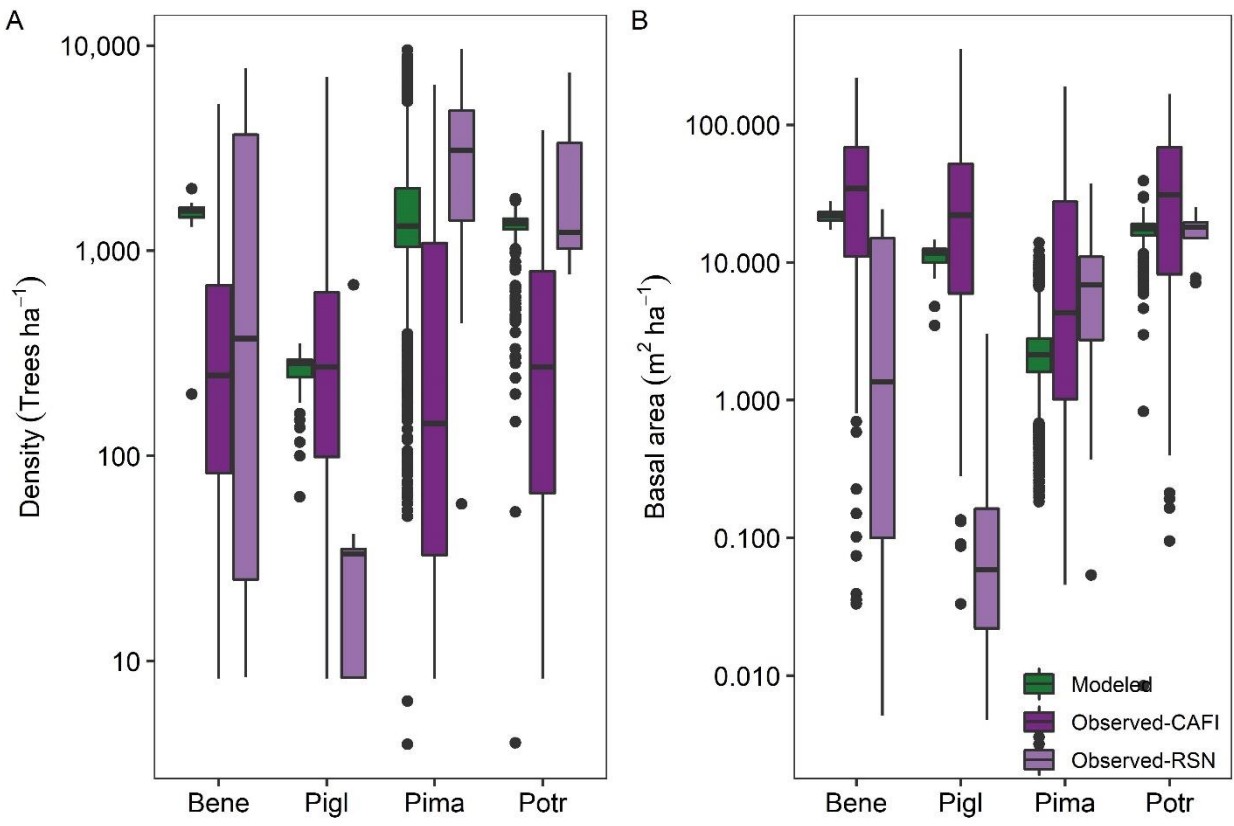

**Figure 8. Simulated and observed stand density and basal area broken out by dominant forest type in a 61,000 ha forested landscape of interior Alaska. Model output is from simulation year 300. Observations are from field sampling in other boreal forest stands (see main text for sources). Pima (*Picea mariana*) = black spruce, Pigl (*Picea glauca*) = white spruce, Potr (*Populus tremuloides*) = trembling aspen, Bene (*Betula neoalaskana*) = Alaskan birch.**



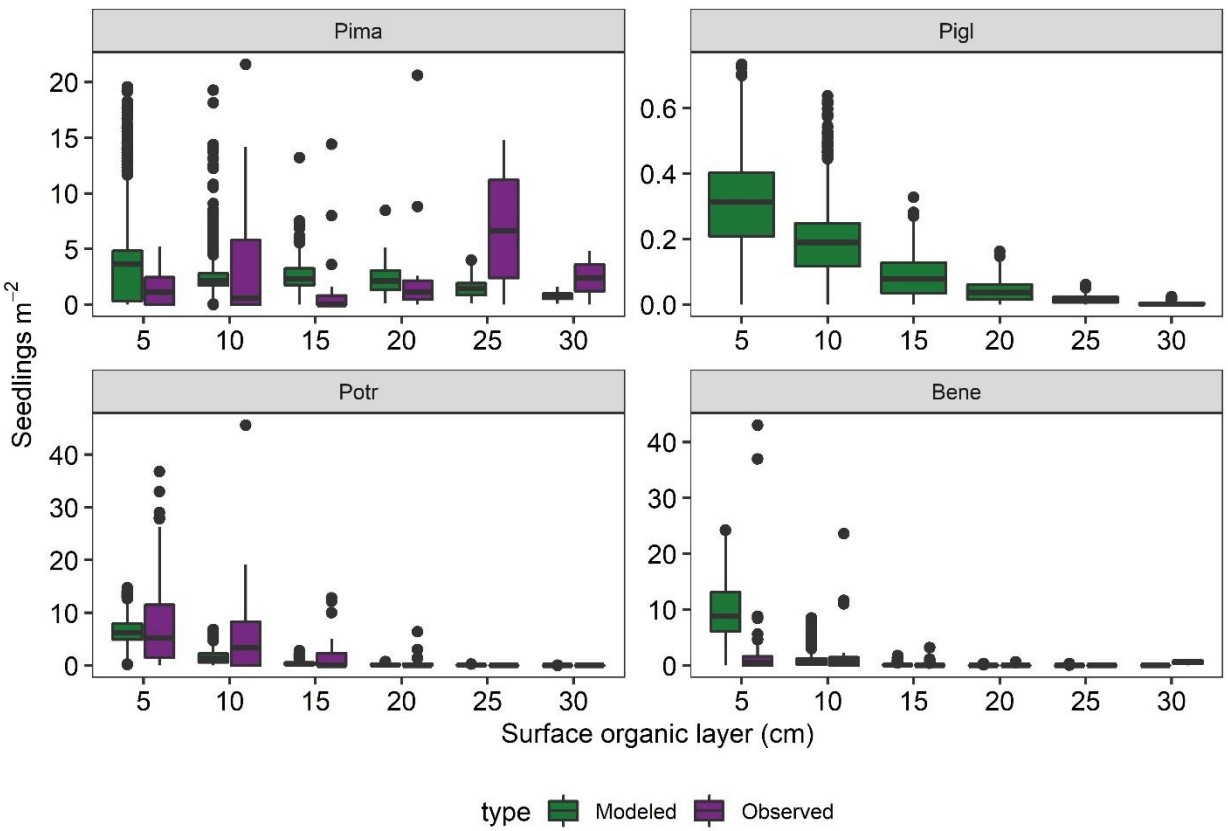

**Figure 9. Simulated and observed tree-seedling density two years postfire as a function of surface organic layer depth. Model output is from recently burned areas in simulation years 260-300 in a 61,000 ha forested landscape in interior Alaska. Observations are from field sampling in other boreal forest stands (see main text for sources). Pima (*Picea mariana*) = black spruce, Pigl (*Picea glauca*) = white spruce, Potr (*Populus tremuloides*) = trembling aspen, Bene (*Betula neoalaskana*) = Alaskan birch.**