# Peer review of "The Permafrost and Organic LayEr module for Forest Models (POLE-FM) 1.0"

_EGUsphere, 2022_

## Author Comment (AC1)

*(Note: The reviewer's comments are in black and the author's responses are in blue. Unless specified otherwise, the line numbers quoted in our responses are with reference to the revised manuscript with track changes turned on.)*

**General comments**

Hansen et al. present a computationally efficient permafrost and soil organic layer module and its coupling to an established forest landscape model. The new module can be used to simulate the annual soil-surface organic layer accumulation and the interannual and seasonal patterns of snow accumulation and active layer depth. Coupled with iLand, the model is used to simulate moss biomass, fire activity, forest composition, stand structure, the soil-surface organic layer accumulation, and the permafrost distribution in a complex landscape in interior Alaska. The computational efficiency of the new module offers great opportunities for the simulation of large spatial extents – also demonstrated here.

We thank the reviewer for the positive assessment of the manuscript.

The manuscript in its current form is well-written and well-structured but requires improvements. It will then be an exciting addition to a growing body of work concerned with the complex interactions between boreal forests and permafrost. As such, some of the model results fit well with observations (e.g. the permafrost distribution). Still, it is unclearly described where exactly these observations were made and how the data was selected from the previous studies cited. The authors should include the most important data and study site descriptions rather than pointing the readers at former studies.

We thank the reviewer for this suggestion. We have added details from benchmarking products. See specific comment on this below for a more detailed response.

To me, some of the figures seem to be suggesting relationships that aren't necessarily statistically proven.

We have revised language throughout the results and discussion section to ensure we are objective in our comparisons to benchmarking products. In particular, we revised lines 352-366. We also revised the discussion section to include more caveats and to explain where we found poor correspondence and what may underpin divergence between model results and benchmarks (lines 435-461).

Adding to this, the time periods for the comparison between observed and simulated data are different between different values (e.g. annual fire probability is compared for the model output years 201-300, and near-surface permafrost for the years 271-300). The reason for this should be discussed.

We wanted to ensure that we were capturing the landscape distribution of permafrost that was consistent over time and not locations that might be transiently considered permafrost. We have revised the window for permafrost to be from model years 261-300 and now clarify in lines 303-305 that a 40-year window (year 261-200) was selected as a multi-decadal period that aligns with

the period used to calculate SOL combustion and postfire seedling density. The time periods for calculating SOL combustion and postfire tree seedling density were chosen to ensure a sufficient sample of fires for analysis while being cognizant of the computational intensity of these calculations. We now state this in lines 318-320. We kept the period of simulation year 201-300 for evaluating the fire regime because fires in Alaska are large and infrequent leading to greater stochasticity in outcomes. We decided at least a century was necessary as it aligns with the mean historical fire return interval in interior Alaska (Johnstone et al. 2010). This is now described in lines 291-292.

The modelling approach is reasonable and I don't see any major issues with the new modules. Nevertheless, the modelled active layer thaw and freezing are delayed compared to observations, and the snow cover height does not fit the observations very well (from what is shown). This raises the question if the modelling results regarding the snow phenology are meaningful. While this can be explained by the fact that the snow parameterization is simple and does not include canopy interception, snow compaction or redistribution, the snow depth is one of the key factors impacting the thermal and hydrological permafrost regime. Similarly, shading and the reduction of wind underneath the forest canopy have been found to severely impact the thermal and hydrological regime of the ground, dampening the below-ground temperatures and changing the available plant water. I understand, that this is not the main focus here but should be mentioned and "used" to explain the found differences between model simulations and observations.

We now include a paragraph in the discussion section (lines 429-451) that describes a number of potential reasons why at times simulated timing of maximum thaw depth and maximum snow depth diverged from observations. We point to specific mechanisms that could be better represented in the model in the future. While not a perfect representation of snow, we are confident the model sufficiently captures these patterns to meet our modeling objectives of representing tree demographics and growth in a high latitude forest landscape

It would additionally be interesting to show the differences and especially the improvements that the new module adds to the overall model performance in terms of stand structure and tree species composition. This could be done by showing an additional map.

We have added lines 401 – 403 to describe differences by running the model again for 300 years with the permafrost and SOL module turned off. We have added such a map as figure 8.

**Specific comments**

l.34: Here it should be noted that while air temperatures are rising (warming) another important aspect are changes in the precipitation patterns (leading to droughts -> increasing fire/pest risks)

We thank the reviewer for mentioning precipitation and its effects on drought. We revised lines 34-36 as follows: "The boreal forest is warming at a rate at least twice the global average (IPCC 2021, Chylek et al. 2022), which can reduce fuel moisture and cause climate-sensitive disturbances, like forest fire, to increase…" While most GCMs project areas underlain by permafrost to dry over the next century (Andresen et al. 2020), precipitation is projected to increase markedly in interior Alaska. Further, past studies have shown historical trends in

gridded precipitation data for Alaska vary in direction and magnitude (McAfee et al. 2014). Thus, the effects of climate change on the hydrologic cycle are of critical importance but extremely uncertain, and as a result, we decided not to mention precipitation trends explicitly.

l.56: This is the case for Alaska/Canada but not in Eastern Siberia. Specify the geographical focus here already.

We have revised line 55 to 56 by specifying in the boreal forests of North America

l.58: Also, hydrology (strongly interlinked with topography).

We now mention hydrology in lines 59-60.

l.59: Furthermore, boreal forests are important in protecting permafrost (shading, snow cover interception, lowering of turbulent heat fluxes, litter layer):

- Chang et al., 2015, Arctic Antarct. Alpine Res. 47 267–79, https://doi.org/10.1657/AAAR00C-14-016
- Fisher et al., 2016, Glob. Change Biol. 22 3127–40, https://doi.org/10.1111/gcb.13248
- Stuenzi et al., 2021, Environ. Res. Lett. 16 084045, https://doi.org/10.1088/1748-9326/ac153d

We thank the reviewer for these great resources. Because our model does not explicitly represent the effects of forest structure on microclimate under the canopy, we opted to not mention these feedbacks here. We do now, however, include further description of the effects of boreal forest on permafrost as mediated by microclimate in the discussion section (lines 440-444).

l.60/61: active-layer vs. active layer

Fixed here and throughout

l.70: It's unclear to me what's meant here in regard to Kruse et al. 2022

We revised lines 70-72 to clarify that Kruse et al. 2022 describes a permafrost module that runs with a fine temporal resolution.

l.73: This is true but there are many newer models available:

- Karra, S., Painter, S. L., and Lichtner, P. C.: Three-phase numerical model for subsurface hydrology in permafrost-affected regions (PFLOTRAN-ICE v1.0), The Cryosphere, 8, 1935–1950, https://doi.org/10.5194/tc-8-1935-2014, 2014.
- Perreault et al. (2021) Numerical modelling of permafrost dynamics under climate change and evolving ground surface conditions: application to an instrumented permafrost mound at Umiujaq, Nunavik (Québec), Canada, Écoscience, 28:3-4, 377-397, https://doi.org/10.1080/11956860.2021.1949819

- Yokohata et al.: Model improvement and future projection of permafrost processes in a global land surface model, Progress in Earth and Planetary Science (2020): https://doi.org/10.1186/s40645-020-00380-w
- Westermann et al.: Simulating the thermal regime and thaw processes of ice-rich permafrost ground with the land-surface model CryoGrid 3, Geosci. Model Dev., 9, 523–546, https://doi.org/10.5194/gmd-9-523-2016, 2016.

We have added these citations to lines 73-76.

l.98: Is this the density of falling snow or the snowpack below the canopy? What about the representation of transient internal snow properties (such as e.g. snow compaction)?

We revised lines 99-100 to clarify it is the density of the snow pack. Our model of snow is simple and does not represent processes such as snow compaction as described in lines 438-446. We have expanded and revised this description in the manuscript to also describe how forest structure can mediate microclimate and snow pack dynamics.

l.124: Air temperature at what height? Above the canopy? There are no canopy density-specific parameterizations/factors for shading of the ground, precipitation interception, or changes in the turbulent surface fluxes, correct? While these are very important factors for permafrost conditions under boreal forest canopies, this is not the focus here. I would, nevertheless, suggest discussing this in more detail in the conclusions section (and possibly restructuring it into a discussion and a short conclusion section).

Air temperature comes from the gridded 1-km climate data. We thank the Reviewer for these suggestions for the discussion. We have discussion of how forest structure and composition mediates microclimate in lines 440-444.

l.147: Why 0.7?

We revised this line. The value is actually 0.92 and was iteratively derived.

l.149: Where is the SLA value for moss from?

We now cite Foster et al. 2019.

l.172: I wasn't able to find this value in the given reference.

We calculated the value from the field observations described in Walker et al. 2020. We have revised lines 177-178 to clarify.

l.176: full full

Fixed

l.180: I'd propose mentioning the reference here.

We have added reference to Seidl et al. 2012 to lines 184-185.

l.236: Where were these measurements conducted? I'd suggest describing the validation studies in more detail here rather than having the readers go through all the individual studies cited.

We have revised section 3 model benchmarking to provide more details for the benchmarking datasets used. For example, in lines 231-233 we write: "To evaluate whether the module could generate realistic intra- and inter-annual patterns of snow accumulation and active layer depth, we selected 17 forested sites in interior Alaska that span approximately 700 km. the southernmost site sits along the Alaskan highway at the border between Canada and Alaska. The northernmost site is just south of the Brooks mountain range along the Dalton highway." We also revised lines 236-238, 243-245, 263-266, 273-275, and 289-291 to provide more detail for other datasets.

l.246: This is unclear: How many are located in mid-elevations, and what is the bias in low- or high-elevation locations?

We have revised lines 252-255 as follows: "When compared with a meteorological station network (SNOTEL), the gridded observational product had a RMSE of 0.32 m with a bias of -0.09 m in mid-elevations (400-800m) where 70% many of our forested sites were located, and a bias of 0.01m at low elevations (< 400m) where the rest of our sites were located (Yi et al., 2020)."

l.340: The model struggles to simulate the timing of thawing and freezing, 15 days is a rather long period compared to the overall short unfrozen season. I suppose this is mostly due to the o

This comment from the Reviewer was not complete. However, in response to feedback from the other Reviewer we have edited lines 352 to 366 in the revised manuscript by removing the use of RMSE. Instead, we present medians and IQRs for observed and simulated variables to be consistent with the rest of the results sections. Thus, we now do not interpret quality of fit as we did before and instead more objectively provide results for the reader to assess. We also further elaborate on caveats and potential explanations in the discussion (lines 429-451).

l.353: close

We have revised lines 375-376 as follows: "The model simulated 39.3% of forested area in the landscape as underlain by permafrost between years 261-300; compared to the estimated of 33.4% of forested area from the benchmarking product." Thus, we now do not interpret quality of fit as we did before, and instead, objectively provide results for the reader to assess

l.398: While this is one of the major advances of the presented model, I'd suggest adding this to the main discussion and omit from adding an extra subchapter for one sentence only.

We believe the reviewer was referring to section 4.5 on line 389 of the original ms. It is no longer included as a separate section and now the sentence can be found on lines 391-392.

l.390: This conclusion reads more like a discussion section, and could maybe be split up into a more extensive discussion and a short conclusion.

We have revised to include both a discussion section and a conclusions section.

l.406: Also, because there is no mechanistic snow parameterization included.

We have expanded our description of our simplistic snow parameterization as a caveat in lines 444-447.

Figure 1: Would you need any references here for the vectorized trees or is this a modified photograph?

This is a modified photograph taken by the author.

Figure 2: The shown relationships (solid lines) between simulated and observed values are misleading. The simulated snow depth seems to be overestimated on average. Instead of the solid lines, it would be interesting to see actual correlation statistics here.

We have revised the caption of figure 2 to clarify that black lines show 1 to 1 relationships rather than correlations. "Figure 2. A. Observed vs simulated maximum annual snow depth at 20 sites between 2001-2017 B.  Observed vs simulated day of spring snow melt at 20 sites between 2001-2017. C. Observed vs simulated maximum annual thaw depth at 7 sites underlain by permafrost between 2014-2018 (only site-years with complete observational records are included). D. Observed vs simulated maximum annual freeze depth at 10 sites not underlain by permafrost between 2014-2018 (only site-years with complete observational records are included). Black lines show one-one relationships in all panels" We appreciate the reviewer's suggestion to include correlation statistics, but opted to maintain consistent approaches of medians and IQRs as was done at landscape scales where correlations are not possible.

Figure 3: Here, only one of seven forest stands is shown which leaves the reader wondering whether this one shows the best fit out of the seven validation sites.

We agree with the reviewer that it would be wonderful to show all the sites and years. But this is not feasible given space constraints. We have revised the caption as follows: "Figure 3. A. Example of daily active layer freezing and thawing. Data from 2016 at one of seven forest stands underlain by permafrost. B Example of daily thawing and freezing. Data from 2016 at one of ten forest stands not underlain by permafrost. Solid lines represent snow depth. Dots represent active layer depth or depth of freezing. Grey fill represents frozen soils. Blue fill represents unfrozen soils."

Figure 4: Is the dark grey area the "sampled landscapes"?

We have revised the caption for figure 4 as follows: "Figure 4. Simulated and observed A. annual fire probability and B. mean burned patch size in a 61,000 ha landscape in interior Alaska. Model output is for years 201-300. Observations are from years 1980-2020. The grey

density distribution shows observed values for all sampled 625 61,000 ha landscapes across the boreal domain of interior Alaska. C. Map showing all 625 sampled landscapes as dark grey squares. Red square shows the landscape simulated in iLand."

Works cited

Andresen, C. G., D. M. Lawrence, C. J. Wilson, A. D. McGuire, C. Koven, K. Schaefer, E.

      Jafarov, S. Peng, X. Chen, I. Gouttevin, et al. 2020. Soil moisture and hydrology

      projections of the permafrost region – a model intercomparison. The Cryosphere 14:445–

      459.

Chylek, P., C. Folland, J. D. Klett, M. Wang, N. Hengartner, G. Lesins, and M. K. Dubey. 2022.

      Annual Mean Arctic Amplification 1970–2020: Observed and Simulated by CMIP6

      Climate Models. Geophysical Research Letters 49:e2022GL099371.

IPCC, editor. 2021. Climate Change 2021: The Physical Science Basis. Contribution of Working

      Group I to the Sixth Assessment Report of the Intergovernmental Panel on Climate

      Change. Cambridge University Press.

Johnstone, J. F., F. S. Chapin, T. N. Hollingsworth, M. C. Mack, V. Romanovsky, and M.

      Turetsky. 2010. Fire, climate change, and forest resilience in interior Alaska. Canadian

      Journal of Forest Research 40:1302–1312.

McAfee, S., G. Guentchev, and J. Eischeid. 2014. Reconciling precipitation trends in Alaska: 2.

      Gridded data analyses. Journal of Geophysical Research: Atmospheres 119:13,820-

      13,837.

---

## Author Comment (AC2)

*(Note: The reviewer's comments are in black and the author's responses are in blue. Unless specified otherwise, the line numbers quoted in our responses are with reference to the revised manuscript with track changes turned on.)*

This paper presents a useful addition to the land modeling community, with a well-described model of permafrost that can be coupled to other land models and has code and documentation available. The need for such a model I felt was well-justified and the paper provides detailed model equations and background, describes and demonstrates the model capabilities, and compares a variety of model outputs against observational data. Overall the model appears to produce reasonable results, and, with some modifications, I believe this paper will make an excellent addition to GMD.

We thank the Reviewer for the positive assessment of the manuscript.

My primary concerns are with the performance claims that I believe need additional support from model validation and with the minimal discussion of model limitations. While the model is compared to observational data, several aspects of this could use further clarification. First, it is unclear to me why the years selected are necessarily comparable to the model results and why different years are chosen for different comparisons, which may just speak to a need for some additional justification for this in the text.

We have addressed this concern by explaining why periods for analysis were chosen and by making the periods more consistent when possible. Please see a more detailed response below to the specific comment about this.

Additionally, differences between model results and data are quantified frequently in terms of RMSE but the reader has no real perspective to understand the relative significance of these RMSE values. This could be addressed by including the range or uncertainty of the observational and simulated values and by giving some measure of statistical significance.

We thank the Reviewer for this suggestion. We have removed the use of RMSE from section 4.1 (lines 352 to 366) and instead present medians and IQRs for observed and simulated variables to be consistent with the rest of the results sections. We chose to not use a measure of statistical significance because sample sizes can be increased with simulations to artificially inflate statistical significance derived from parametric approaches (Lucash et al. 2019). We now describe this in lines 346-347.

Also, some discrepancies between model results and observations seem to not really be discussed or have only limited discussion, such as the model's near-surface permafrost extent by aspect being considerably worse in the north aspect than others, later maximum annual active-layer depth or earlier freeze depth. It would be helpful to address these more and help the reader understand potential causes and the implications of these for interpreting other model results.

We now include two paragraphs in the discussion section (lines 429-451 and 452-461) that describes a number of potential reasons why at times simulated timing of maximum thaw depth and maximum snow depth diverged from observations. We also describe discrepancies between

simulated and observed permafrost distributions with aspect at the landscape scale. We point to specific underpinning mechanisms that could be better represented in the model in the future.

Model limitations (such as critical assumptions or missing processes and their implications for results) are only lightly touched on in the conclusions section, and I believe the paper would benefit from further elaboration on these points, to help the reader put this model in perspective and better understand how to consider its results.

Please see response to comment above.

In the interest of putting this model in perspective, it would additionally be useful to see some comparison of iLand's performance with and without this new module to some of the observational datasets to demonstrate the value of this module beyond providing additional outputs.

We have added lines 401 – 403 to describe differences by running the model again for 300 years with the permafrost and SOL turned off. We have added such a map as figure 8.

**Specific comments:**

Equation 9 - Would be helpful to add units to the constant to make this conversion clearer to the reader

We have revised line 125 to include units for the constant in equation 9

L.280 Unclear to me why we expect these model simulation years to correspond to these data years. I suggest adding some clarification on this, here and in other places where model results are compared to time-specific data.

We wanted to ensure that we were capturing the landscape distribution of permafrost that was consistent over time and not locations that might be transiently considered permafrost. We have revised the window for permafrost to be for 261-300 and clarify in lines 303-305 that a 40-year window (year 261-200) was selected as a multi-decadal period that aligns with the period used to calculate SOL combustion and postfire seedling density. The time periods for calculating SOL combustion and postfire tree seedling density were chosen to ensure a sufficient sample of fires for analysis while being cognizant of the computational intensity of these calculations. We now state this in lines 318-320. We kept the period of simulation year 201-300 for evaluating the fire regime because fires in Alaska are large and infrequent leading to greater stochasticity in outcomes. We decided at least a century was necessary as it aligns with the mean historical fire return interval in interior Alaska (Johnstone et al. 2010). This is now described in lines 291-292.

L.353-354 Somewhat unclear. Would be helpful to elaborate more on the importance of aspect and mention the differences in results between north versus other aspects.

We have revised lines 377-378 as follows: "Simulated permafrost was over represented on north-facing slopes, as compared to the benchmarking product, but corresponded well on all other aspects."

L.375 Would be useful to add comparison here to iLand results without the permafrost module

We have added lines 401 – 403 to describe differences by running the model again for 300 years with the permafrost and SOL module turned off. We have added such a map as figure 8

L.397-398 The claim "Benchmarking results demonstrate the model recreates temporal and spatial patterns consistent with observations" feels not completely supported by the text and I believe needs more justification or caveats.

We have revised lines 426-428 as follows: "With some exceptions discussed below, benchmarking results generally demonstrate the model recreates temporal and spatial patterns consistent with observations at stand to landscape scales over days to centuries." In the discussion section, we further address caveats including the simplicity of the snow model, not representing effects of forest structure on microclimate and consequences for snow and permafrost, and the effects of tree species composition on landscape patterns of permafrost distribution in lines 429-461.

L.398-399 The claim "Our model will contribute to improving 21st -century projections of boreal forest change." is not really supported by the current results, but would be by the addition of a figure or data showing improvements to iLand's projections with the addition of the permafrost module.

We have added such a map as figure 8.

Figure 3. The representation of frozen and unfrozen soil in this figure is somewhat confusing, as it seems like there should be two different fills - one for observed, one for simulated - but instead there is only the one. Can this be separated out or otherwise clarified?

We thank the reviewer for pointing this out. We have revised the caption of figure 3 by adding "Grey fill represents simulated frozen soils. Blue fill represents simulated unfrozen soils." (lines 851-852).

Figure 5. Maybe add the years for the observed dataset in this figure caption, since the simulation years are given.

We added the years for the benchmarking product (1990-2013) to the figure caption in lines 864-865.

Figure 7. Some clarification on the caption for Panel B would be helpful, as it took me a minute to figure out what was meant here.

We have revised lines 876-879 as follows: "Figure 7. A. Tree-species composition in a 61,000 ha forested landscape of interior Alaska used to initialize iLand. B. Changes in tree species dominance over 300 years of simulation. Pima (*Picea mariana*) = black spruce, Pigl (*Picea glauca*) = white spruce, Potr (*Populus tremuloides*) = trembling aspen, Bene (*Betula neoalaskana*) = Alaskan birch"

Works cited

Johnstone, J. F., F. S. Chapin, T. N. Hollingsworth, M. C. Mack, V. Romanovsky, and M. Turetsky. 2010. Fire, climate change, and forest resilience in interior Alaska. Canadian Journal of Forest Research 40:1302–1312.

Lucash, M. S., K. L. Ruckert, R. E. Nicholas, R. M. Scheller, and E. A. H. Smithwick. 2019. Complex interactions among successional trajectories and climate govern spatial resilience after severe windstorms in central Wisconsin, USA. Landscape Ecology.